# C-Reactive Protein Velocity (CRPv) as a New Biomarker for the Early Detection of Acute Infection/Inflammation

**DOI:** 10.3390/ijms23158100

**Published:** 2022-07-22

**Authors:** Tal Levinson, Asaf Wasserman

**Affiliations:** 1Department of Internal Medicine “E”, Tel Aviv Sourasky Medical Center, and Sackler Faculty of Medicine, Tel Aviv University, Tel Aviv 6423906, Israel; asafw@tlvmc.gov.il; 2Infectious Diseases Unit, Tel Aviv Sourasky Medical Center, and Sackler Faculty of Medicine, Tel Aviv University, Tel Aviv 6423906, Israel

**Keywords:** C-reactive protein, velocity, inflammation, infection, bacterial, viral

## Abstract

C-reactive protein (CRP) is considered a biomarker of infection/inflammation. It is a commonly used tool for early detection of infection in the emergency room or as a point-of-care test and especially for differentiating between bacterial and viral infections, affecting decisions of admission and initiation of antibiotic treatments. As C-reactive protein is part of a dynamic and continuous inflammatory process, a single CRP measurement, especially at low concentrations, may erroneously lead to a wrong classification of an infection as viral over bacterial and delay appropriate antibiotic treatment. In the present review, we introduce the concept of C-reactive protein dynamics, measuring the velocity of C-reactive protein elevation, as a tool to increase this biomarker’s diagnostic ability. We review the studies that helped define new metrics such as estimated C-reactive protein velocity (velocity of C-reactive protein elevation from symptoms’ onset to first C-reactive protein measurement) and the measured C-reactive protein velocity (velocity between sequential C-reactive protein measurements) and the use of these metrics in different clinical scenarios. We also discuss future research directions for this novel metric.

## 1. Introduction

C-reactive protein (CRP) is an inflammatory biomarker and is one of the downstream mediators of the acute-phase response [1]. CRP is synthesized by the liver in response to the secretion of several inflammatory cytokines including interleukin (IL)-1, IL-6, and tumor necrosis factor (TNF). These proinflammatory cytokines increase the concentration of CRP and support the ongoing inflammatory process, while a decrease in their concentration usually heralds the subsiding and termination of inflammation [2]. Certain CRP isoforms activate the complement pathway, induce phagocytosis, and promote apoptosis, while different isoforms promote the chemotaxis and recruitment of circulating leukocytes to areas of inflammation and can delay apoptosis [3]. CRP’s main role in inflammation is the activation of the C1q molecule in the complement pathway, leading to the opsonization of pathogens, hence actively participating in the immune response to infection [4].

CRP is commonly used by clinicians in acute bacterial diseases for both the detection of the inflammatory process and for the quantization of its intensity [5,6,7,8]. Furthermore, CRP is used to guide antibiotic treatment [9,10,11,12] and for the identification of the resolution of the inflammatory process [13]. In fact, acute bacterial infections have been repeatedly associated with increased CRP concentrations, and this parameter is generally used by clinicians to clarify whether a certain patient presents a significant inflammatory response or not [14,15]. While most emergency room physicians expect an elevated CRP level during acute bacterial infections, a first low CRP may result in an underestimation of the severity of the eventual septic conditions and lead to the erroneous assessment of the severity of the underlying inflammatory process [16,17].

## 2. Defining Early CRP Dynamics

As a single CRP measurement has limited efficacy in the differential diagnosis between acute bacterial and viral infections, we defined two terms to investigate the early dynamics of CRP over time. Estimated CRP velocity (eCRPv) is defined as the level of the first CRP measurement divided by the time from the patient’s first reported symptom (e.g., fever) to the CRP being measured and expressed as the velocity of CRP measured in mg/L/h.

The idea behind this term is that two theoretical patients presenting with the same level of CRP, one having been ill for a few hours and the other for a few days, probably represent a different inflammatory response. For example, if the two patients present to the emergency room with the same CRP levels of 100 mg/mL, where one has been sick for 10 h and the other for 100 h, the eCRPv will be significantly different, 10 mg/L/h and 1 mg/L/h, respectively. It seems likely that the patient who presents with the more rapid rise in CRP, meaning a higher eCRPv, has a higher probability of developing a cytokine storm [18]. 

CRP velocity (CRPv) upon admission is defined by the dynamics of the first two CRP measurements from admission (CRP1 to CRP2) divided by the time (in hours) between the two tests, wheras eCRPv has the advantage of describing the dynamics of CRP from the beginning of the disease process until the arrival at the emergency room, However, eCRPv depends on the subjective estimation of disease duration, while CRPv can be more accurately calculated and usually represents the first 24 h in the hospital stay. The conclusions of studies regarding the utility of CRP kinetic properties for both infectious and noninfectious inflammatory processes are summarized in Table 1 and Table 2. 

## 3. Using CRP Dynamics in the Diagnosis of Infectious Diseases

CRP is commonly used in the evaluation of patients presenting to the emergency room with an acute febrile disease. In that scenario, patients with an elevated CRP are more likely to be considered suffering from a bacterial infection and therefore be admitted and started on early antibiotic treatment, while patients with a low CRP may be more easily discharged or have a delay in treatment. However, there is a substantial range of CRP values that correlate with bacterial as well as with viral infections, and therefore cannot be relied upon to differentiate between these two types of infectious etiologies. Furthermore, few studies have shown that a single CRP test without consideration of this inflammatory biomarker’s kinetics might convey an erroneous impression of a relatively mild infection.

Paran et al. [19] reviewed 173 patients arriving with fever at an emergency department. Patients diagnosed with a bacterial infection had a median CRP of 63 mg/L and CRPv of 3.61 mg/L/h, and patients with a non-bacterial diagnosis had a median CRP of 23 mg/L and CRPv of 0.41 mg/L/hour. CRPv improved differentiation between the bacterial and non-bacterial febrile patients compared with CRP alone. Of interest was the subset of patients who presented to the emergency room with low CRP but eventually were found to have a bacterial infection, which manifested the potential disadvantage of a single CRP measurement without consideration of this biomarker’s kinetic properties when encountering a patient with bacterial infection and developing inflammation. 

A retrospective cohort [20] examined 2284 patients presenting with sepsis to the emergency room. The authors were able to identify 175 patients (7.6%) who, despite meeting criteria for sepsis, had a low, apparently normal (<31 mg/L) first CRP, and this patient group had an unfavorable outcome of 19.4% mortality within a week, mostly due to bacterial sepsis (pneumonia and urinary tract infections). In that cohort, there was a significant increase in median CRP within the first 24 h of hospitalization, from a median CRP of 16.1 mg/L (IQR 7.9–22.5) to 58.6 mg/L (IQR 24.2–134.4), *p* < 0.001, and this was more pronouncedly demonstrated with a change in CRP velocity from a CRPv of 0.4 ± 0.29 mg/L/h to a CRPv of 8.3 ± 24.2 mg/L/h (*p* < 0.001).

In a similar cohort [21] of 643 patients who were hospitalized with a relatively low first CRP (<60 mg/dL) and eventually had a definitive bacterial or viral infection, patients with bacterial infections had a first CRP measurement, which was higher than that of viral patients, but had limited ability to differentiate between the groups. Using a second CRP and CRPv, the diagnostic accuracy was increased from an area under the curve (AUC) of 0.57 for CRP1 to an AUC of 0.77 and 0.83 for CRP2 and CRPv, respectively. The authors were able to create a cutoff of CRP velocity of 3.47 mg/dL/h as 93.8% specific and 50.2% sensitive for the diagnosis of a bacterial over viral infection. It should be noted that in the group of patients presenting with low CRP and a bacterial infection, there was a significant difference between the eCRPv and the CRPv. This was also demonstrated in a cohort [22] of patients presenting to the emergency room with Gram-negative bacteremia. Of 2200 patients with bacteremia, 460 had a low first CRP (<30 mg/L) of whom 229 were further investigated to find that they had a significant five-fold higher C-reactive protein level with their second test. This notion was further strengthened by Bernstein et al. [23], who demonstrated that, in a cohort of 136 patients who presented to the emergency department with first CRP ≤ 31.9 mg/L, a second measurement of CRP value within 24 h of admission enabled the calculation of the CRPv, which was much higher in patients with acute bacterial infections compared to those with acute viral infection (CRPv 4.4 ± 2.7 and 0.9 ± 1.2 respectively, *p* < 0.001). Furthermore, when calculating the eCRPv, its value was greater in patients with bacterial infections compared to patients with viral infections (1.2 ± 1.1 and 0.8 ± 1.6 respectively, *p*-value < 0.001, AUC 0.7, CI 0.62–0.77). 

In an additional recently described retrospective cohort of patients admitted to the departments of internal medicine with apparently normal CRP concentration, a short-term follow-up CRP test within twenty-four hours of admission was performed, in order to determine the relation between 7-day mortality and these CRP values [24]. Among 3504 inpatients, the mean first and second measurements of CRP were 8.8 (8.5) and 14.6 (21.6) mg/L, respectively. The authors divided the first and the second CRP results into quartiles according to the CRP concentration with increasing CRP concentration in each consecutive quartile. The seven-day mortality rates increased from 1.7% in the first CRP quartile to 7.8% in the fourth one (*p* < 0.0001). With regard to the second CRP, the seven-day mortality rates increased according to the CRP increment being 0.5% in the first quartile as opposed to 9.5% in the fourth one (*p* < 0.0001). Hence, while the death percentage was 4.6 times higher in the fourth as opposed to the first quartile of the first CRP test, this difference was 19 times higher in the fourth as opposed to the first quartile of the follow-up CRP test. The AUC of the ROC curve, when using the first CRP measurement as the predictor of 7-day mortality, was 0.639 (0.599–0.680), *p* < 0.001. This AUC increased to 0.731 (0.696–0.766), *p* < 0.001 when using the second measurement of CRP. Interestingly, it should be noted that in the above-described cohort of patients, the sepsis cause of death increased in a dose-dependent manner with the quartiles of the first and second CRP. Patients with an extremely low level of CRP (first quartile) not only had a better survival rate, but also had a lower risk of mortality from sepsis compared to patients in the highest quartile of either the first or second measurement of CRP.

All of these studies manifest that despite presenting with a relatively low-grade inflammatory response that could potentially be observed in an apparently healthy population, these patients might harbor severe and potentially lethal medical conditions, which might be overlooked by the treating physician upon clinical presentation in light of the apparently normal first CRP measurement. Further strengthening this concept was a recent study of patients admitted with a very low C-reactive protein concentration [25]. These findings strongly suggested not relying on a single apparently normal CRP concentration upon admission to a medical facility but insisting on at least one, if not more than one, additional test to follow.

This spike in CRP velocity cannot be explained just by the natural course of bacterial infections, and a possible explanation for this dynamic is the administration of antibiotics between the first and second CRP tests. Patients with low CRP despite having a bacterial infection may suffer from immune stunning, which is reduced due the effect of antibiotics on the bacterial load, or alternatively, the destruction of bacteria and release of bacterial endotoxins causes a laboratory “Jarich–Herxheimer”-like reaction leading to immune activation.

Justo et al. [26] tried to exploit this possible response to antibiotic treatment as a tool to differentiate between community-acquired pneumonia (CAP) and chronic-obstructive lung disease (COPD) exacerbation. CRP levels on the day of admission and prior to antibiotic administration were higher in CAP patients than in COPD patients but with significant overlap. Following the administration of antibiotics, the second CRP made a sharp increase of 36.7% of CAP patients compared to only 5.9% of COPD exacerbation patients (*p* = 0.005) and remained unchanged in 61.8% of COPD patients compared to 16.3% of CAP patients (*p* = 0.0006). 

Similarly, CRP kinetics were used by Povoa et al. to predict ventilator-associated pneumonia (VAP) [27]. All 35 microbiologically documented VAP cases were assessed by the kinetics of CRP from day 1 to day 6 of therapy. CRP kinetics and its relative changes were significantly different between survivors and non-survivors of VAP (*p* = 0.026 and *p* = 0.005, respectively), whereas the kinetic properties of the biomarkers procalcitonin and the mid-region fragment of pro-adrenomedullin did not distinguish between survivors and non-survivors. Hence, CRP kinetics after prescription of antibiotics therapy is useful in the identification of VAP patients with poor outcome and performs better than other biomarkers.

Yet, on a small-scale observational retrospective study by Pereira et al. [28], 60 critically ill patients with community-acquired pneumonia, aspiration pneumonia, and bacteremia admitted to the intensive care unit had their CRP levels collected consecutively for up to eight days and were followed up to one year. No significant association was found between CRP kinetics and early or late mortality and antibiotic treatment duration (*p* > 0.05).

If proven on a larger scale, CRP’s response to antibiotic administration may be used both for the diagnosis of bacterial infections and to confirm the appropriateness of an empiric treatment, an extremely important question in the age of resistant bacteria and antibiotics overuse, therefore potentially allowing for better antibiotic stewardship.

The kinetics of CRP were shown to be clinically useful to identify patients with poor outcome after community-acquired blood stream infection (CA-BSI) and to predict short- and long-term mortality up to a year. In a population-based study by Povoa et al. [29], 935 patients had their CRP levels measured on day 1 and day 4 of CA-BSI, and the relative CRP variation in relation to day 1 CRP was evaluated and defined as CRP ratio. At day 4, CRP level decreased in patients who survived to day 365 and in patients who died on day 4 to day 30, and in patients who died on day 31 to day 365; however, at day 4, the CRP ratio was lower in survivors on day 365 when compared to non-survivors of day 4 to day 30 (*p* < 0.001) and of day 31 to day 365 (*p* < 0.001). Persistent inflammation measured as early as day 4, as assessed by CRP ratio, was strongly correlated with patient mortality.

The utility of the kinetic properties of CRP was shown to improve the diagnostic ability of postsurgical infections from non-infectious inflammation. Nahum et al. found that in children undergoing cardiac surgery, the usage of CRPv could assist in detecting bacterial infection and differentiating it from non-infectious systemic inflammatory reaction such as the reaction secondary to the bypass procedure itself [30]. In total, 121 children who underwent cardiac surgery with bypass were tested for CRP up to five days postoperatively and during febrile episodes. A comparison was made between patients with proven bacterial infection, febrile patients without bacterial infection and non-febrile patients. CRPv was significantly higher in the infectious group (4 ± 4.2 mg/dL per day) than in the fever-only group (0.6 ± 1.6 mg/dL per day, *p* < 0.001). CRPv of 4 mg/dL/day had a positive predictive value of 85.7% for bacterial infection with 95.2% specificity.

## 4. Using CRP Dynamics in Non-Infectious, Inflammatory Disorders

The ischemic injury and myocardial necrosis following ST elevation myocardial infarction (STEMI) incite an acute inflammatory response. Holzknecht et al. sought to characterize the pathophysiological process linking CRPv and myocardial infarction pathology [31]. In their prospective cohort study of 316 patients with STEMI undergoing primary percutaneous intervention (PCI), the patients were examined with cardiac magnetic resonance in order to evaluate their microvascular obstruction and its association with CRPv. The study demonstrated significant association between CRPv and the occurrence of microvascular obstruction even after adjustment to the cardiac troponin level and TIMI flow (odds ratio 2.7, 95% confidence interval 1.54–4.73; *p* = 0.001). In addition, CRPv was found to be a better predictor of microvascular obstruction compared to 24 h CRP (AUC difference: 0.03, *p* = 0.002). The authors concluded that CRPv could be used as an early and sensitive biomarker for more severe infarct pathology and outcome among STEMI patients undergoing PCI, and potentially, CRPv could help to identify patients who might benefit from anti-inflammatory and cardio protective treatment.

The inflammatory response in STEMI is not confined solely to the infarct zone due to up-regulation of cytokine expression in the non-infarcted myocardium [32]. The elevation of inflammatory markers, and CRP specifically, are associated with adverse clinical outcomes including recurrent ischemia, heart failure and mortality. The optimal timing, however, for measuring CRP following a coronary event has not been determined. Milwidsky et al. [33] sought to explore the use of early CRP dynamics in a cohort of patients admitted due to acute STEMI. In a retrospective analysis of 492 consecutive patients with STEMI who underwent early invasive intervention and had two CRP measurements taken within the first 24 h of hospitalization, the second CRP and CRPv were significantly higher among patients who died within 30 days of admission, and CRPv was an independent predictor of 30-day mortality. The first CRP, however, was not associated with increased mortality, stressing the need for sequential tests.

When studying early left ventricular dysfunction in patients with first STEMI and its association with CRPv, Holzknecht et al. evaluated 432 STEMI patients who underwent cardiac magnetic resonance imaging at a median of 3 days after primary PCI in order to determine the left ventricular function and the characteristics of the myocardial infarction [34]. The CRPv was independently associated with the left ventricular ejection fraction (LVEF) (*p* = 0.004) and LVEF ≤ 40% (odds ratio 1.71, 95% confidence interval 1.19–2.45, *p* = 0.004), meaning that CRPv was independently associated with early left ventricular dysfunction after STEMI treated with primary PCI.

The association between left ventricular function in patients with STEMI according to echocardiographic parameters and CRPv was studied by Banai et al. [35]. A cohort of 1059 patients admitted due to STEMI and treated with primary PCI were examined by echocardiography. Patients with high CRPv had lower LVEF, and CRPv was found to independently predict LVEF ≤ 35% (hazard ratio 1.3, confidence interval 95% 1.21–1.4, *p* < 0.001) and grade III diastolic dysfunction (hazard ratio 1.16, confidence interval 95% 1.02–1.31, *p* = 0.02). The kinetics of CRP expressed as CRPv had a better diagnostic profile for severe systolic dysfunction compared to CRP (AUC 0.734 ± 0.02 vs. 0.608 ± 0.02). In conclusion, CRPv was found to be a predictive marker for both systolic and diastolic dysfunction in STEMI patients treated with primary PCI. 

Moreover, in a recent study that examined the pathophysiological effect of CRP itself as a mediator of tissue damage in acute myocardial infarction, Ries et al. explored the usage of CRP apheresis in patients aiming to investigate the relationship between CRP gradient and the myocardial infarct size and function in the setting of acute STEMI as well as the effect of CRP apheresis efficacy [36]. A total of 45 apheresis patients and 38 controls were recruited. CRP apheresis was performed 24 ± 12, 48 ± 12, and optionally 72 ± 12 h after onset of symptoms with a median CRP concentration of 23.0 mg/L (range 9–279) at first apheresis. Mean CRP depletion achieved over all apheresis procedures was 53.0 ± 15.1%. Apheresis sessions were well-tolerated. Reduced infarct size in the apheresis group compared to the control group (primary endpoint) was not achieved; however, three major adverse cardiac events occurred in the control group after 12 months, but none occurred in the apheresis group. The authors concluded that CRP concentrations could effectively be reduced by CRP apheresis, which has the potential to interfere with deleterious aspects of STEMI and warrants a larger randomized study.

Zahler et al. showed that among 801 STEMI patient who underwent PCI and had CRPv calculated within 24 h after admission, patients who developed new onset atrial fibrillation following PCI had a significantly higher median CRPv (1.27 vs. 0.43 mg/L/h, *p* = 0.002) [37]. The authors concluded that CRPv might be an independent biomarker associated with increased risk for new-onset atrial fibrillation in this group of patients. The ability to early detect these inflammation-prone patients may allow implementing an anti-inflammatory intervention to decrease the risk for unfavorable outcomes associated with atrial fibrillation in this group of patients.

Zahler et al. [38] also examined the association between early CRP velocity and acute kidney injury (AKI) in STEMI patients undergoing primary percutaneous intervention. As periprocedural elevation in CRP is associated with increased kidney injury, it was shown that CRPv was independently associated with kidney injury and patients with CRP velocity above 0.8 mg/L/h having a fourfold higher risk for AKI (15.2% vs. 3.8%, *p* < 0.01). Additionally, for each 0.1 mg/L/h increase in CRPv, the risk for AKI increased by 3%.

## 5. Future Directions for CRP Dynamics

This review focused on the velocity of rise in CRP at the beginning of an inflammatory/infectious process. In everyday practice, clinicians also examine the decline in CRP levels as a tool to follow on the response to treatment and recovery of a patient.

Multiple studies [10,39,40] have attempted to set thresholds for declining levels of CRP to guide the duration of antibiotics treatment, the switch to oral antibiotics, and safe discharge; however, these studies focused on specific levels of CRP and not on the rate of CRP decline. We found no studies that examined an early CRP decline velocity as a tool for identification of response to antibiotic treatment.

The world is currently facing a global pandemic due to severe acute respiratory coronavirus 2 (SARS-CoV-2 virus), which is the causative agent of coronavirus disease 2019 (COVID-19). This new disease brought to the forefront the concept of cytokine storm [18,41], as the disease is thought to have a so-called early “viral phase” and a late “inflammatory phase”, with immunosuppressive medications being prescribed and investigated for this later phase.

CRP and other cytokines have been widely studied in COVID-19, with correlation to a more severe disease, lung damage, and mortality [9,42]. Even when compared to other inflammatory biomarkers, CRP concentration at admission to the hospital of COVID-19 patients correlated with disease severity and tended to be a good predictor of adverse outcomes [43]. Whether CRP velocity, as described in this review, can help in identifying the transition to the inflammatory phase is an open question, deserving further investigation.

There are several readily used clinical implications to the kinetic approach when confronting patients with acute infection/inflammation and relatively low CRP concentrations. Clinicians in the department of emergency medicine should not discharge those patients without repeating the CRP test, hence making sure that the patients do not harbor an inflammatory burst. Cardiologists could use this biomarker as a signal to stop nephrotoxic drugs in patients with acute myocardial infarction before the development of acute kidney injury as well as starting cardioprotective medications before cardiac failure is evident. Patients with significantly elevated CRP velocities should be hospitalized and monitored carefully, and, in case bacterial infections are suspected, they should be given antibiotic treatment as soon as possible. Special care should be provided to those patients at the extremes of CRP velocities.

In summary, C-reactive protein is a commonly used biomarker for detecting and differentiating between bacterial and viral infections. A single CRP measurement, especially when low, can mislead the clinicians to rule out a bacterial infection. In this review, we presented the dynamic features of CRP and the advantage of measuring its rate of rise in order to ameliorate its diagnostic ability.

## Figures and Tables

**Table 1 ijms-23-08100-t001:** CRP velocity in infectious disorders.

Paran et al., 2009	CRPv improved differentiation between bacterial and non-bacterial infections
Nahum et al., 2012	CRPv improves the diagnostic ability of postsurgical infections from non-infectious inflammation following cardiac surgery
Povoa et al., 2016	CRP kinetics differentiates between survivors and non-survivors of ventilator associated pneumonia
Wasserman et al., 2019	Septic patients with first low CRP had increased CRPv within 24 h of hospitalization
Pereira et al., 2019	No significant association between CRP kinetics and early or late mortality and antibiotic treatment duration in patients with pneumonia
Coster et al., 2020	CRPv increased the diagnostic accuracy between bacterial and viral infections in hospitalized patients
Povoa et al., 2020	CRP kinetics is useful in identifying patients with poor outcome after community acquired blood stream infection and predict short- and long-term mortality up to a year
Bernstein et al., 2021	CRPv is significantly higher in patients with acute bacterial infections compared to acute viral infection in patients presenting with first low CRP (≤31.9 mg/L)

CRP—C-reactive protein. CRPv—C-reactive protein velocity.

**Table 2 ijms-23-08100-t002:** CRP velocity in noninfectious inflammatory disorders.

Milwidsky et al., 2017	CRPv is an independent predictor of 30-day mortality in patients with STEMI
Zahler et al., 2019	CRPv is independently associated with acute kidney injury in STEMI patients
Holzknecht et al., 2021	CRPv is independently associated with early left ventricular dysfunction following STEMI
Holzknecht et al., 2021	CRPv is significantly associated with microvascular obstruction in STEMI patients
Zahler et al., 2021	CRPv is significantly higher in STEMI patients who develop new onset atrial fibrillation
Banai et al., 2022	CRPv is predictive for both systolic and diastolic dysfunction in STEMI patients

CRPv—C-reactive protein velocity. STEMI—ST elevation myocardial infarction.

## Data Availability

The data that support the findings of the study are available from the corresponding author upon reasonable request.

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
