# Peer review of "C-Reactive Protein Velocity (CRPv) as a New Biomarker for the Early Detection of Acute Infection/Inflammation"

_ijms, 2022, doi:10.3390/ijms23158100_

Round 1

Reviewer 1 Report

Although present review is interesting and provides significant information on CRPs, however it lack in depth insights and discussion on its implications. It is suggested that authors provide elaborate discussions by citing proper examples.

Moreover, inclusion of figures/schemes illustrating mechanism of CRPs and its biological function would greatly improve quality of the work and increase readability of the manuscript.

Minor comments:

Table alignments could be redone.

The referencing style is incorrect.

Headlines should be informative, can be elaborated.

There are several too short paragraphs which do not convey significant information or are meaningless. For e.g., L47-49; L61-63; and L303-304 (there are more such paras). A para should provide complete information about 1 or 2 points.    

I suggest authors address these issues and resubmit.

Reviewer 2 Report

The authors of this manuscript (Manuscript ID: ijms-1802009) propose that the kinetic properties of CRP can be used as a new biomarker for the early detection of acute infection/inflammation. In the manuscript, there is an adequate coverage of the existing studies regarding the utility of C-reactive protein velocity for the diagnosis of infectious diseases, distinguishing between acute viral and bacterial infections, and the usage of CRP dynamics in early diagnosis of non-infectious, inflammatory disorders. The manuscript is interesting, well organized and highlights the up-to date difficulty for physicians in using a single CRP measurement for the diagnosis of bacterial infections with relatively low CRP concentrations.

Below are some minor comments that need to be addressed.

1.       Please rephrase sentence 104-107

2.       Please briefly describe the findings of the study cited as ref 25 (line 152)

Reviewer 3 Report

This is a very important review about C - reactive protein (CRP) and CRP dynamics in inflammation, very useful in clinical practice.

CRP is synthesized by the liver in response to the secretion PRO -inflammatory cytokines CRP elevation is frequent in acute bacterial inflammation, but in practice there are patients with a first low CRP level, but having a severe bacterial inflammation.

The authors then propose to evaluate the CRP dynamics as a more sensitive biomarker, using 2 parameters:

-          estimated C-reactive 18 protein velocity (velocity of C-reactive protein elevation from symptoms onset to first C-reactive  protein measurement) and

-          measured C-reactive protein velocity (velocity between sequential C-reactive protein measurements)

C-reactive protein is a commonly used biomarker for detecting and differentiating between bacterial and viral infections.

Evaluating the dynamics instead of simply a single CRP determination help clinicians to better diagnose a bacterial infection and to administer early the antibiotics needed.

The review clear, comprehensive and on a very important issue of everyday practice.

Author Response

There are only two reviewers of our manuscript.

Round 2

Reviewer 1 Report

The authors have addressed most of the comments.

This manuscript is a resubmission of an earlier submission. The following is a list of the peer review reports and author responses from that submission.